# Determinants of early mental health help-seeking among women in Bangladesh: A nationally representative bootstrapped regression analysis

Tanvir Ahmed[1]*, Salma Tasnim Luthfa[2], Amatul Haque Chaahat[3],
Masudur Rahman Kanchon[4], Azaz Bin Sharif[5]

1 Climate Change, Environment and Health, BRAC James P Grant School of Public Health, Dhaka, Bangladesh, 2 Department of Public Health, North South University, Dhaka, Bangladesh, 3 Department of Public Health, BRAC James P Grant School of Public Health, Dhaka, Bangladesh, 4 Center of Excellence for Health Systems and Universal Health Coverage, BRAC James P Grant School of Public Health, Dhaka, Bangladesh, 5 Department of Public Health, North South University, Dhaka, Bangladesh

☺ These Authors contributed equally to this work.

* dr.tanviirahmed@gmail.com

## Abstract

Mental health is a critical public health concern, shaping emotional, psychological, and social well-being. Depression and anxiety are the most common and preventable mental disorders, can be significantly improved by early interventions and proper care. This study investigates help-seeking behavior at early stage (mild to moderate anxiety and depression) and its determinants among Bangladeshi women. Special focus is given to media exposure as a potential enabler of early support. We used Bangladesh Demographic and Health Survey (BDHS) data 2022 and utilized bootstrapped fixed-effect logistic regression model to detect the determinants. Only 20.8% of the women in this study (Weighted N = 3181) reported seeking help at least once. Compared to women with no education, those with secondary or higher education were respectively 1.41 and 1.55 times more likely to seek help (95% CI = 1.07 – 1.87 and 1.04 – 2.28 respectively)). In reference to Barisal, residents of Khulna (aOR = 2.76, 95% CI = 1.70-4.65) showed the highest increase in help-seeking odds. Working women had significantly greater odds of seeking help (aOR = 2.17, 95% CI: 1.52–3.07). Internet users were twice as likely to seek help (95% CI: 1.52–2.75), while smartphone users showed lower odds (aOR = 0.64, 95% CI: 0.46–0.88). Watching Television showed no significant (aOR = 1.08, 95% CI = 0.88, 1.36) impact on help-seeking behaviour. There are alarmingly low levels of help-seeking among Bangladeshi women facing early-stage anxiety and depression, with housewives being the least likely to seek support. Education, employment, decision making autonomy, and internet usage significantly influenced help-seeking behavior. Despite widespread access, television remains a neglected tool in mental health promotion. An integrated, culturally sensitive strategy with combination of mass media,

**Data availability statement:** Original Dataset, RScript for extracting test-ready dataset and the csv format of test ready dataset can be found here: https://doi.org/10.6084/m9.figshare.29834708.v1.

**Funding:** The authors received no specific funding for this work.

**Competing interests:** The authors have declared that no competing interests exist.

**Abbreviations:** FY, Fiscal Year; BDHS, Bangladesh Demographic and Household Survey; PHQ, Patient Health Questionnaire; GAD, Generalized Anxiety Disorder; LMIC, Low Middle Income Country; LDC, Least Developed Country.

peer support, equitable responsibilities, education, and policy have the potential to empower women and spark early help-seeking for mental health.

## Introduction

Mental health encompasses emotional, psychological, and social well-being, influencing thought processes, emotions, behaviors, stress management, relationships, and decision-making. Recognizing its significance, the World Health Organization (WHO) has prioritized mental health for accelerated action in its Thirteenth General Programme of Work (GPW13) [1]. Depression and anxiety are the most common mental disorders. Both are treatable, and in many cases, preventable. Early intervention, proper care, and support can significantly improve outcomes and enhance overall well-being [2,3]. Slowing down the progression of or averting these disorders at early stage can help reduce the overall mental health disease burden, and facilitate an effective & complete recovery and reduce the economic costs linked to anxiety & depressive disorders [4–6]. In Bangladesh, approximately 18.7% of adults and 12.6% of children experience mental health disorders, with depression, anxiety, and stress being the most prevalent [8,9]. The situation assessment report by WHO special initiative of mental health estimated that 92.3% of adults with mental disorders nationwide do not receive any treatment [7]. Bangladesh's mental healthcare system remains critically inadequate due to a shortage of public facilities, limited skilled professionals, insufficient funding, weak policy implementation, and the influence of stigma and cultural beliefs [9,14]. For many individuals, particularly those in rural areas, reaching a tertiary facility is not only financially burdensome but also logistically impractical. By the time individuals seek help at tertiary centers, their conditions are often severe, requiring extensive interventions that could have been avoided with earlier support [7,10].

Culture influences anxiety & depressive symptoms, awareness, and care-seeking, with more somatic symptoms reported in non-Western countries [11,12]. Studies found higher somatic symptoms among South Asians than Caucasians leading to recognize that sociocultural factors interact with risk factors, shaping anxiety and depression's manifestation [11,13]. Women are disproportionately affected due to a combination of biological, hormonal, social, and cultural factors, a disparity that is particularly pronounced in LMICs [14–17]. In Bangladesh, Symptoms of Anxiety and Depression are largely unrecognized, suggesting potential underreporting due to unfamiliar terminology [11]. The underestimation have serious consequences, leading to low prioritization of women's mental health in family health planning and inadequate interventions [18,19].

In Bangladesh, television programs such as *Meena Cartoon* [20,21], along with public service advertisements addressing the dangers of smoking, substance abuse, tuberculosis, and HIV, have already demonstrated success in driving significant social transformation. These types of initiatives have effectively raised awareness, shifted attitudes, and promoted healthier behaviors across diverse populations not only in

Bangladesh but also in other LMICs as well [22–28]. Popular formats like dramas, cartoons, music videos, and live shows have successfully driven behavior change in various health domains [29,30]. As of 2025, our desk review revealed that not a single dedicated mainstream TV program, advertisement, or campaign in state-owned television channel targets mental health awareness, early symptom recognition, or help-seeking behavior, also only 3 of the 40 private television channels broadcast dedicated programs for mental health but views of these programs on Youtube is very low suggesting very poor reach to mass population.

## Aims and objectives

As women are at heightened risk of from severe mental health burdens [31–34], this study aims to determine the prevalence of the practice of help-seeking behavior among women experiencing mild to moderate Anxiety and Depression in Bangladesh and identify its proximal determinants with special focus on impact of Media Exposure on their help-seeking behavior. The study recognizes that understanding the help-seeking patterns in this demographic is crucial to designing effective mental health interventions specific to women's needs in the Bangladeshi context. We assessed whether Smartphone, Internet, Newspaper/Magazine, Radio and Television exposure correlate with a greater likelihood of engaging in help-seeking behavior and their nature (positive/negative) of influence on help seeking.

## Methods

We conducted a secondary analysis on data from 'Individual Recode (BDIR81)' file from the 2022 Bangladesh Demographic and Health Survey (BDHS), which includes comprehensive data on women's characteristics collected using the women's questionnaire along with selected household-level variables. We've also utilized the GIS file (BDGC81FL) to access location of all 674 clusters. The BDHS utilizes a two-stage stratified probability sampling design for ensuring national representativeness in the sample. At first stage, the country is stratified by administrative divisions and residence type (urban/rural). Then, within each stratum, clusters of households are selected using probability proportional to the size of each stratum, followed by systematic sampling of households within each cluster. Further details on the survey methodology are available on the DHS website [35,36].

## Ethical statement

This study is based on a secondary analysis of data obtained from the Demographic and Health Surveys (DHS) Program. Access to the dataset was granted by DHS after the submission and approval of a formal data request, in full compliance with their data use policies and procedures. The dataset used is fully anonymized and contains no personally identifiable information. Moreover, there exists no mechanism by which individuals can be re-identified from the data. As the analysis involves secondary data that is publicly available upon approved request and does not involve direct interaction with human subjects, this study was exempt from further institutional ethical review. But all ethical principles outlined in the Declaration of Helsinki and guidelines for responsible conduct in secondary data use were upheld throughout the study. We affirm that the data were used exclusively for scholarly purposes as permitted under the DHS data use agreement.

## Study population

Individual recode file of BDHS 2022 contains data of ever-married women of reproductive age (15–49 years) in Bangladesh. We have included the women who reported being in early stage of anxiety and depression, had PHQ and GAD score between 5–14 (mild to moderate Depression and anxiety, stages at which active treatment is not commonly recommended, further screening is required for pharmacotherapy [37,38]). Out of 30078 women, 3211 (unweighted) women were selected for our study who met the inclusion criteria. (Fig 1)

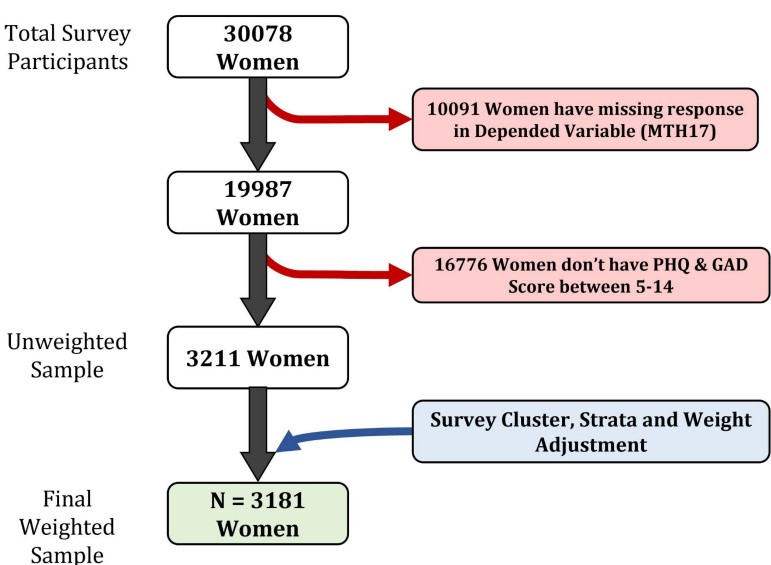

**Fig 1. Study population inclusion and exclusion criteria.**

## Study measures

We covered individual characteristics of the participants (Age, Education Level, Occupation, For whom they work for, Marital Status and Religion), Husband/Partner's characteristics (Education Level, Occupation, Currently residing with wife/ partner), Household characteristics (Wealth Index, No. of members in the household, Residence and Division), Media Exposure variables (Smartphone ownership, Internet usage, Frequency of Reading newspaper, listening to radio and watching television) and Decision making authority of the participants about Respondent's Healthcare as the explanatory variables of outcome measure which is addressed by participant's one of the dichotomous responses to 'Ever tried to seek help for the things you experienced' variable which points towards the experiences mentioned in PHQ-9 and GAD-7 questionnaires. Survey related variable: Primary Sampling Unit, Stratification used in sample design and Women's individual sample weight were also kept into the test-ready data set for incorporating survey design into our analysis. Cases having missing responses for outcome variable or many missing responses in explanatory variables were dropped from our study.

## Data management & analysis

We used R for Windows (v4.4.1) and RStudio (v2024.09.1 build 394) for data wrangling and analyses, Spatial Distribution was determined using QGis (v3.42 Munster). Sociodemographic variables, media exposure, and women's decision-making autonomy related variables were recoded and releveled to align our study with existing literature. Weight variable was generated from women's individual sample weight. Before analysis, survey design was taken into account by utilizing the primary sampling units, stratification used in sample design and weight to ensure correct and generalizable estimation. After incorporating the survey design, we proceeded with the weighted frequency of our final sample which was 3181. A descriptive analysis was done on all the independent and outcome variables. Spatial distribution of cluster-wise proportion of women who sought help at early stage of anxiety and depression were calculated from matching the cluster numbers from Individual Recode file with GIS data.

Association of the categorical independent variables with outcome were detected using Chi-square test of Independence with cluster design effect adjustment assured with Rao-Scott's adjustment and strength of association of each

independent variables was measured by Cramer's V statistic. Regarding the design of survey and data collection, scope of a hierarchical logistic model was explored but keeping 'Strata' as random intercept did not show improvement in variance over the fixed effect model. So, strata is kept as one of the fixed variables in the model. A univariable and two multivariable weighted univariate fixed effect logistic regression model were employed to estimate crude and confounder-adjusted odds ratios. Between two multivariable logistic models, all the explanatory variables were included into full model to adjust for potential confounders. Given the categorical nature of all variables, absence of multicollinearity, and our focus on explanation rather than prediction, we preferred AIC-based 'Backward' stepwise elimination method over other regularization methods for variable selection to prevent overfitting of the model. Metrices of the final model are as follows, AIC = 2911, McFadden's Pseudo $R^2$ = 0.23, Logarithmic Loss = 0.38, PCP = 69.3%. Assumptions of the model were met (S1 Fig) and performance metrices (AUC = 0.67, Accuracy = 0.79, Brier Score = 0.14, F1 Score = 0.88) are suggestive of a good performing model and Hosmer-Lemeshow Goodness of Fit test (Chi-square = 7.005, df = 8, p = 0.536) also assures a well fitted model. We fitted the model with non-parametric bootstrapped (1000 resamples) datasets for robust estimation of the average point (odds ratios) and interval (2.5th and 97.5th Percentiles of odds ratios) estimates. Bootstrapping involved repeatedly resampling the data to generate more stable and robust confidence intervals. More information about the analysis procedure is provided in the supporting document (S1 File).

## Results

This study enrolled a total of 3181 participants (Table 1). The mean age of participants was 33.8 years (SD = 8.8), with the highest proportion (19.1%) in the 35–39 age group. Most participants (42.3%) had secondary education and 60.4% were housewives. Majority of the women resided in rural region (71%). Not all women who owned a smartphone were internet users. In the domain of media exposure, proportion of smartphone users and internet users remained notably low, with less than one-fourth (27% and 24% respectively) women had reported usage. Traditional media engagement was even more limited with less than 5% of women reported listening to the radio or reading newspapers or magazines, a decline likely driven by the rise of digital technologies. However, over half of the women (55%) reported watching television regularly. Only 15.6% women made independent healthcare decisions, while 60.7% decided jointly with others. Only 20.8% of the participants had ever sought help at early stage of symptoms (Fig 2). Prevalence of early help seeking in majority of the clusters fell below 5% (S2 Fig)

The results of the chi-square tests (Table 2) showed association between a number of socio-demographic characteristics and help seeking behavior. Employment status & the nature of employment were key factors (p < 0.001). Employed (25.5%) & self-employed (32.0%) individuals were more likely to seek help behavior. Compared to the participants with younger partners (16.8%), those having older partners (25.9%) were found more associated to seek support. Internet users (26.0%) were significantly more likely to seek mental health support than non-users (19.1%). Among divisions, Barishal had the lowest (13.1%) and Khulna had the highest proportion (31.3%) of help seeking women. (Fig 3). Decision-making autonomy in respondent's healthcare, large household purchase, visit to family or relatives and spending husband/ partner's earning were significantly (p < 0.001) associated with seeking help for mental health issues where women who are independent decision makers were more likely to seek help for their situation.

From logistic regression analysis (Table 3), Education level of the participants, their husband/partner's age, residence and division are found to be significant predictors (Fig 4). Though working status of the participants is not significant, for whom the working for is a significant factor influencing help seeking behavior at early stage of anxiety and depression. Decision-making autonomy and exposure to media showed a mixed association with women's help-seeking behavior. Greater autonomy in own healthcare decision making and smartphone usage appeared to hinder health-seeking, while in Internet usage is found to facilitate service seeking.

Compared to women with no formal education, women who completed secondary or Higher education has shown 41% (aOR = 1.41, 95% CI = 1.07-1.87) and 55% (aOR = 1.55, 95% CI = 1.04-2.28) higher odds of seeking help respectively.

**Table 1. Socio-demographic characteristics of the study participants (n = 3181).**

| Characteristic | n (%) |
|---|---|
| **Age of the Participants** | |
| 15-19 | 196 (6.2%) |
| 20-24 | 352 (11.1%) |
| 25-29 | 502 (15.8%) |
| 30-34 | 572 (18.0%) |
| 35-39 | 608 (19.1%) |
| 40-44 | 494 (15.5%) |
| 45-49 | 458 (14.4%) |
| **Education Level of the Participants** | |
| No education | 593 (18.6%) |
| Primary | 927 (29.1%) |
| Secondary | 1,346 (42.3%) |
| Higher | 315 (9.9%) |
| **Employment Status of the Participant** | |
| Unemployed | 1,924 (60.5%) |
| Employed | 1,256 (39.5%) |
| **Participants Working for** | |
| Housewife | 1,923 (60.4%) |
| Family member | 1,080 (33.9%) |
| Self Employed/For Others | 178 (5.6%) |
| **Marital Status of the Participant** | |
| Married | 2,902 (91.3%) |
| Widowed, Divorced or Separated | 278 (8.7%) |
| **Religion** | |
| Islam | 2,940 (92.4%) |
| Hinduism & Others | 241 (7.6%) |
| **Age Difference with Husband** | |
| 0-5 Years | 1016 (35%) |
| 6-10 Years | 1056 (36.4%) |
| More than 10 Years | 831 (28.6%) |
| **Husband/Partner's Education Level** | |
| No education | 889 (27.9%) |
| Primary | 919 (28.9%) |
| Secondary | 980 (30.8%) |
| Higher | 393 (12.4%) |
| **Husband/Partner's Occupation** | |
| Unemployed | 144 (4.5%) |
| Employed | 3,036 (95.5%) |
| **Living Arrangement with Wife** | |
| Living Together | 2,655 (83.5%) |
| Living elsewhere | 526 (16.5%) |
| **Sex of the Household Head** | |
| Female | 503 (15.8%) |
| Male | 2,677 (84.2%) |
| **Relationship with Household Head** | |
| Head Herself | 373 (11.7%) |

*(Continued)*

| Characteristic | n (%) |
|---|---|
| Wife of Head | 2,071 (65.1%) |
| Daughter/Daughter in law/Granddaughter | 575 (18.1%) |
| Mother/Mother-in-Law/Others | 161 (5.1%) |
| **Number of Members in Household** | |
| Up to 4 | 1,632 (51.3%) |
| 4+ | 1,549 (48.7%) |
| **Wealth Index** | |
| Poorest | 669 (21.0%) |
| Middle | 652 (20.5%) |
| Poorer | 700 (22.0%) |
| Richer | 630 (19.8%) |
| Richest | 530 (16.7%) |
| **Residence** | |
| Rural | 2,314 (72.8%) |
| Urban | 866 (27.2%) |
| **Division** | |
| Barishal | 178 (5.6%) |
| Chattogram | 624 (19.6%) |
| Dhaka | 685 (21.5%) |
| Khulna | 390 (12.3%) |
| Mymensingh | 208 (6.6%) |
| Rajshahi | 437 (13.7%) |
| Rangpur | 488 (15.3%) |
| Sylhet | 171 (5.4%) |
| **Smartphone Usage** | |
| Doesn't have any | 1,046 (32.9%) |
| Not a Smartphone | 1,260 (39.6%) |
| Have a Smartphone | 875 (27.5%) |
| **Internet Usage** | |
| Never | 2,411 (75.8%) |
| Uses Internet | 769 (24.2%) |
| **Frequency of reading Newspaper/Magazine** | |
| Not at all | 3,000 (94.3%) |
| Watches sometimes/ regularly | 180 (5.7%) |
| **Frequency of listening to Radio** | |
| Not at all | 3,091 (97.2%) |
| Watches sometimes/ regularly | 90 (2.8%) |
| **Frequency of Watching Television** | |
| Not at all | 1,433 (45.1%) |
| Watches sometimes/ regularly | 1,747 (54.9%) |
| **Decision Maker of Respondent's Healthcare** | |
| Independently | 496 (15.6%) |
| Jointly | 1,929 (60.7%) |
| Others | 756 (23.8%) |

*(Continued)*

**Table 1.** (Continued)

| Characteristic | n (%) |
|---|---|
| **Decision Maker of Large Household Purchase** | |
| Independently | 301 (9.5%) |
| Jointly | 1,925 (60.5%) |
| Others | 955 (30.0%) |
| **Decision Maker of Visiting to Family/Relative** | |
| Independently | 396 (12.4%) |
| Jointly | 2,053 (64.5%) |
| Others | 732 (23.0%) |
| **Decision Maker of spending Husband/Partner's Earning** | |
| No Earning | 53 (1.7%) |
| Independently | 205 (6.5%) |
| Jointly | 2,005 (63.0%) |
| Others | 917 (28.8%) |
| **Ever sought help for the things Participant experienced** | |
| Yes | 660 (20.8%) |
| No | 2521 (79.2%) |

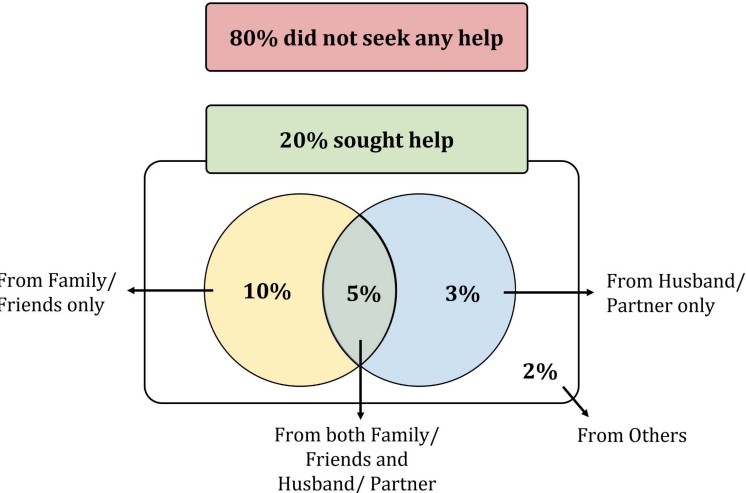

**Fig 2. Proportion of the study population who sought help for mild to moderate anxiety and depression.**

Women whose age gap with husband is between 6–10 years and more than 10 years have respectively 28% (aOR = 1.28, 95% CI = 1.09-1.61) and 69% (aOR = 1.69, 95% CI = 1.34-2.14) increase in of help seeking over women whose age gap with husband is 0–5 years. Urban women have lower odds (aOR = 0.76, 95% CI = 0.61, 0.94) compared to Rural women. In contrast to Barishal, all other divisions have shown higher odds of help seeking but estimates from only Chattogram (aOR = 2.18, 95% CI = 1.36-3.62), Khulna (aOR = 2.76, 95% CI = 1.70-4.65) and Rajshahi (aOR = 1.71, 95% CI = 1.05-2.90) are statistically significant. Employed women who are either working for family members (aOR = 1.47, 95%

**Table 2. Association between socio-demographic characteristics & ever sought help for the things they've experienced with cramer's v statistic for significant variables. (n = 3181).**

| Characteristic | No N = 2,521[1] | Yes N = 660[1] | p-value[1] |
|---|---|---|---|
| **Age of the Participants** | | | 0.10 |
| 15-19 | 168 (85.9%) | 28 (14.1%) | |
| 20-24 | 288 (81.8%) | 64 (18.2%) | |
| 25-29 | 408 (81.3%) | 94 (18.7%) | |
| 30-34 | 450 (78.8%) | 121 (21.2%) | |
| 35-39 | 474 (77.9%) | 135 (22.1%) | |
| 40-44 | 387 (78.3%) | 107 (21.7%) | |
| 45-49 | 346 (75.5%) | 112 (24.5%) | |
| **Education Level of the Participants** | | | 0.5 |
| No education | 479 (80.9%) | 114 (19.1%) | |
| Primary | 744 (80.3%) | 183 (19.7%) | |
| Secondary | 1,054 (78.3%) | 292 (21.7%) | |
| Higher | 243 (77.1%) | 72 (22.9%) | |
| **Employment Status of the Participant (Cramer's V = 0.096)** | | | **<0.001** |
| Unemployed | 1,585 (82.3%) | 340 (17.7%) | |
| Employed | 936 (74.5%) | 321 (25.5%) | |
| **Participants Working for (Cramer's V = 0.105)** | | | **<0.001** |
| Housewife | 1,583 (82.3%) | 340 (17.7%) | |
| Family member | 816 (75.6%) | 264 (24.4%) | |
| Self Employed/For Others | 121 (68.0%) | 57 (32.0%) | |
| **Marital Status of the Participant** | | | 0.8 |
| Married | 2,298 (79.2%) | 604 (20.8%) | |
| Widowed, Divorced or Separated | 222 (79.8%) | 56 (20.2%) | |
| **Religion** | | | 0.5 |
| Islam | 2,334 (79.4%) | 605 (20.6%) | |
| Hinduism & Others | 186 (77.2%) | 55 (22.8%) | |
| **Age Difference with Husband (Cramer's V = 0.084)** | | | **<0.001** |
| 0-5 Years | 845 (83.2%) | 171 (16.8%) | |
| 6-10 Years | 838 (79.4%) | 218 (20.6%) | |
| More than 10 Years | 615 (74.1%) | 215 (25.9%) | |
| **Husband/Partner's Education Level** | | | 0.3 |
| No education | 721 (81.1%) | 168 (18.9%) | |
| Primary | 734 (79.9%) | 185 (20.1%) | |
| Secondary | 766 (78.1%) | 214 (21.9%) | |
| Higher | 300 (76.3%) | 93 (23.7%) | |
| **Husband/Partner's Occupation** | | | 0.2 |
| Unemployed | 108 (74.5%) | 37 (25.5%) | |
| Employed | 2,413 (79.5%) | 624 (20.5%) | |
| **Living Arrangement with Wife (Cramer's V = 0.029)** | | | **0.043** |
| Living Together | 2,122 (79.9%) | 533 (20.1%) | |
| Living elsewhere | 398 (75.7%) | 128 (24.3%) | |
| **Sex of the Household Head** | | | 0.2 |
| Female | 386 (76.6%) | 118 (23.4%) | |
| Male | 2,135 (79.7%) | 543 (20.3%) | |

*(Continued)*

**Table 2.** (Continued)

| Characteristic | No<br>N = 2,521[1] | Yes<br>N = 660[1] | p-value[1] |
|---|---|---|---|
| **Relationship with Household Head** | | | 0.15 |
| Head Herself | 283 (75.8%) | 90 (24.2%) | |
| Wife of Head | 1,647 (79.6%) | 423 (20.4%) | |
| Daughter/Daughter in law/Granddaughter | 469 (81.6%) | 106 (18.4%) | |
| Mother/Mother-in-Law/Others | 121 (74.9%) | 41 (25.1%) | |
| **Number of Members in Household** | | | 0.8 |
| Up to 4 | 1,289 (79.0%) | 343 (21.0%) | |
| 4+ | 1,231 (79.5%) | 318 (20.5%) | |
| **Wealth Index** | | | 0.4 |
| Poorest | 546 (81.7%) | 122 (18.3%) | |
| Middle | 504 (77.3%) | 148 (22.7%) | |
| Poorer | 564 (80.5%) | 137 (19.5%) | |
| Richer | 491 (78.0%) | 138 (22.0%) | |
| Richest | 415 (78.3%) | 115 (21.7%) | |
| **Residence** | | | 0.15 |
| Rural | 1,815 (78.4%) | 500 (21.6%) | |
| Urban | 706 (81.5%) | 161 (18.5%) | |
| **Division (Cramer's V = 0.137)** | | | **<0.001** |
| Barishal | 155 (86.9%) | 23 (13.1%) | |
| Chattogram | 472 (75.7%) | 151 (24.3%) | |
| Dhaka | 569 (83.1%) | 116 (16.9%) | |
| Khulna | 268 (68.7%) | 122 (31.3%) | |
| Mymensingh | 171 (82.2%) | 37 (17.8%) | |
| Rajshahi | 342 (78.4%) | 94 (21.6%) | |
| Rangpur | 402 (82.4%) | 86 (17.6%) | |
| Sylhet | 141 (82.3%) | 30 (17.7%) | |
| **Smartphone Usage** | | | 0.8 |
| Doesn't have any | 831 (79.5%) | 214 (20.5%) | |
| Not a Smartphone | 1,003 (79.6%) | 257 (20.4%) | |
| Have a Smartphone | 685 (78.4%) | 189 (21.6%) | |
| **Internet Usage (Cramer's V = 0.084)** | | | **<0.001** |
| Never | 1,951 (80.9%) | 460 (19.1%) | |
| Uses Internet | 569 (74.0%) | 200 (26.0%) | |
| **Frequency of reading Newspaper/Magazine** | | | 0.058 |
| Not at all | 2,390 (79.6%) | 611 (20.4%) | |
| Watches sometimes/ regularly | 131 (72.4%) | 50 (27.6%) | |
| **Frequency of listening to Radio** | | | 0.2 |
| Not at all | 2,454 (79.4%) | 637 (20.6%) | |
| Watches sometimes/ regularly | 66 (73.7%) | 24 (26.3%) | |
| **Frequency of Watching Television** | | | 0.2 |
| Not at all | 1,152 (80.3%) | 282 (19.7%) | |
| Watches sometimes/ regularly | 1,369 (78.3%) | 378 (21.7%) | |
| **Decision Maker of Respondent's Healthcare (Cramer's V = 0.127)** | | | **<0.001** |
| Respondent independently | 352 (71.1%) | 143 (28.9%) | |
| Jointly | 1,605 (83.2%) | 324 (16.8%) | |
| Others | 563 (74.4%) | 193 (25.6%) | |

*(Continued)*

**Table 2.** (Continued)

| Characteristic | No<br>N = 2,521[1] | Yes<br>N = 660[1] | p-value[1] |
|---|---|---|---|
| **Decision Maker of Large Household Purchase (Cramer's V = 0.084)** | | | **<0.001** |
| Respondent independently | 210 (69.9%) | 91 (30.1%) | |
| Jointly | 1,567 (81.4%) | 358 (18.6%) | |
| Others | 743 (77.8%) | 212 (22.2%) | |
| **Decision Maker of Visiting to Family/Relative (Cramer's V = 0.117)** | | | **<0.001** |
| Respondent independently | 293 (74.0%) | 103 (26.0%) | |
| Jointly | 1,691 (82.4%) | 362 (17.6%) | |
| Others | 536 (73.3%) | 196 (26.7%) | |
| **Decision Maker of spending Husband/Partner's Earning (Cramer's V = 0.098)** | | | **<0.001** |
| Independently | 184 (71.1%) | 74 (28.7%) | |
| Jointly | 1,638 (81.7%) | 367 (18.3%) | |
| Others | 698 (76.1%) | 220 (23.9%) | |

[1] Chi-Square test of Independence with Rao-Scott's Adjustment for Survey Design.

CI = 1.21-1.79) or self-employed/work for others (aOR = 2.17, 95% CI = 1.52-3.07) have significantly higher odds of help seeking over housewives. There is no significant difference between Non-smartphone and Non users of mobile phone but Smartphone users have 34% lower odds (aOR = 0.64, 95% CI = 0.46-0.88) of help seeking behavior compared to those who doesn't have any mobile phone. Internet users have 2.04 times higher odds (95% CI = 1.52-2.75) of help seeking in contrast to women who never used internet. Frequency of watching television has no significant effect on the outcome. Comparing with sole decision maker women of their own healthcare, those who take decision jointly or whose decision is taken by others has lower odds of help seeking but only joint decision takers have 39% lower (aOR = 0.61, 95% CI = 0.45-0.83) odds which is statistically significant. Women from families where decision of family visits or visit to relative is taken by others have 1.51 times higher odds of seeking help which is statistically significant (95% CI = 1.11-1.89).

## Discussion

Ours is the first ever study to explore such early-stage help-seeking behavior among the female population from nationally representative data with the target to explore proximal determinants and to investigate the current impact of media exposure on help-seeking behavior at mild to moderate anxiety and depressive stage.

We've found that about 8 out of ten women who experienced mild to moderate anxiety and depression didn't seek any kind of help. Multiple studies on women of reproductive age, perinatal period also found extremely low proportion of help seeking women in Bangladesh as well [39–41]. Studies suggest that limited awareness about mental health, widespread stigma around seeking help, and poor service availability, socio-cultural factors—like traditional beliefs, religious and social taboos further discourage care-seeking behavior [39,40]. But respondent's religion didn't showed any significant change in care seeking behavior in our study.

Our study revealed that almost half the women are educated below secondary level but women who are educated higher than that had showed higher tendency of help seeking for their experiences. The association of higher education and positive health is now unquestionable, prior studies established that benefit of higher education on women's health is more impactful than it is on men's [42–46]. Though secondary education or higher showed significant increase in help seeking behavior, national curriculum of textbooks for primary, secondary and higher secondary education contain no information about basic concepts of mental health till 2022 which was a major barrier towards awareness buildup mental wellbeing [7,47]. With support of UNICEF Ministry of Education of Bangladesh has employed a new curriculum for

**Table 3. Estimates from logistic regression model, of the factors influencing help seeking behavior (n = 3181).**

| Model Specification | Unadjusted Model[1] | | Full Model[2] | | Bootstrapped Model[3] | |
|---|---|---|---|---|---|---|
| Characteristic | OR[4,5] | 95% CI[5] | OR[4,5] | 95% CI[5] | OR[4,5] | 95% CI[5] |
| **Age of the Participants** | | | | | | |
| *15-19* | 1.00 | — | 1.00 | — | – | |
| *20-24* | 1.44 | 0.90, 2.36 | 1.38 | 0.83, 2.35 | | |
| *25-29* | 1.45 | 0.92, 2.34 | 1.16 | 0.67, 2.05 | | |
| *30-34* | 1.51 | 0.97, 2.43 | 1.28 | 0.71, 2.35 | | |
| *35-39* | **1.64***  | 1.06, 2.62 | 1.28 | 0.70, 2.38 | | |
| *40-44* | **1.68***  | 1.08, 2.71 | 1.00 | 0.51, 1.95 | | |
| *45-49* | **1.79***  | 1.15, 2.89 | 1.15 | 0.58, 2.30 | | |
| **Education Level of the Participants** | | | | | | |
| *No education* | 1.00 | — | 1.00 | — | 1.00 | — |
| *Primary* | 1.12 | 0.86, 1.46 | 1.11 | 0.83, 1.50 | 1.20 | 0.90, 1.58 |
| *Secondary* | **1.28***  | 1.01, 1.64 | 1.23 | 0.90, 1.68 | **1.41***  | 1.07, 1.87 |
| *Higher* | **1.48***  | 1.07, 2.04 | 1.24 | 0.77, 1.99 | **1.55***  | 1.04, 2.28 |
| **Employment Status of the Participants** | | | | | | |
| *Unemployed* | 1.00 | — | 1.00 | — | – | |
| *Employed* | **1.60*** ** | 1.35, 1.90 | 1.06 | 0.84, 1.12 | | |
| **For Whom Participants Working** | | | | | | |
| *Housewife* | 1.00 | — | 1.00 | — | 1.00 | — |
| *For Family member* | **1.50*** ** | 1.25, 1.80 | **1.41*** ** | 1.21, 1.79 | **1.47*** ** | 1.21, 1.79 |
| *Self Employed/For Others* | **2.22*** ** | 1.59, 3.05 | **2.08*** ** | 1.52, 3.07 | **2.17*** ** | 1.52, 3.07 |
| **Marital Status of the Participants** | | | | | | |
| *Married* | 1.00 | — | 1.00 | — | – | |
| *Widowed, Divorced or Separated* | 0.92 | 0.67, 1.23 | 0.84 | 0.57, 1.22 | | |
| **Religion** | | | | | | |
| *Islam* | 1.00 | — | 1.00 | — | – | |
| *Hinduism & Others* | 0.96 | 0.69, 1.30 | 1.04 | 0.73, 1.44 | | |
| **Age Difference with Husband** | | | | | | |
| *0-5 Years* | 1.00 | — | 1.00 | — | 1.00 | — |
| *6-10 Years* | 1.22 | 0.98, 1.52 | 1.24 | 0.98, 1.56 | **1.28*** * | 1.09, 1.61 |
| *More than 10 Years* | **1.51*** ** | 1.32, 2.25 | **1.64*** ** | 1.46, 3.84 | **1.69*** ** | 1.34, 2.14 |
| **Husband/Partner's Education Level** | | | | | | |
| *No education* | 1.00 | — | 1.00 | — | – | |
| *Higher* | **1.35***  | 1.01, 1.79 | 1.36 | 0.91, 2.03 | | |
| *Primary* | 1.15 | 0.91, 1.45 | 1.17 | 0.90, 1.53 | | |
| *Secondary* | **1.32***  | 1.06, 1.66 | 1.19 | 0.89, 1.57 | | |
| **Husband/Partner's Occupation** | | | | | | |
| *Unemployed* | 1.00 | — | 1.00 | — | – | |
| *Employed* | 0.76 | 0.52, 1.12 | 1.22 | 0.75, 2.06 | | |
| **Living Arrangement with Wife** | | | | | | |
| *Living Together* | 1.00 | — | 1.00 | — | – | |
| *Living elsewhere* | 1.24 | 0.99, 1.54 | 1.10 | 0.77, 1.56 | | |
| **Sex of the Household Head** | | | | | | |
| *Female* | 1.00 | — | 1.00 | — | – | |
| *Male* | 0.84 | 0.67, 1.06 | 0.96 | 0.59, 1.62 | | |

*(Continued)*

**Table 3.** (Continued)

| Model Specification | Unadjusted Model[1] | | Full Model[2] | | Bootstrapped Model[3] | |
|---|---|---|---|---|---|---|
| **Relationship with Household Head** | | | | | | |
| *Head Herself* | 1.00 | — | 1.00 | — | – | |
| *Wife of Head* | 0.83 | 0.64, 1.08 | 1.01 | 0.54, 1.87 | | |
| *Daughter/Daughter in law/Granddaughter* | 0.76 | 0.56, 1.04 | 1.01 | 0.57, 1.76 | | |
| *Mother/Mother-in-Law/Others* | 0.91 | 0.59, 1.39 | 1.41 | 0.74, 2.64 | | |
| **Number of Members in Household** | | | | | | |
| *Up to 4* | 1.00 | — | 1.00 | — | – | |
| *4+* | 0.92 | 0.77, 1.09 | 0.94 | 0.77, 1.15 | | |
| **Wealth Index** | | | | | | |
| *Poorest* | 1.00 | — | 1.00 | — | – | |
| *Middle* | **1.38**[*] | 1.05, 1.82 | 1.20 | 0.89, 1.63 | | |
| *Poorer* | 1.12 | 0.85, 1.47 | 0.97 | 0.73, 1.30 | | |
| *Richer* | **1.46**[**] | 1.11, 1.91 | 1.11 | 0.80, 1.52 | | |
| *Richest* | **1.40**[*] | 1.06, 1.86 | 1.07 | 0.73, 1.57 | | |
| **Residence** | | | | | | |
| *Rural* | 1.00 | — | 1.00 | — | 1.00 | — |
| *Urban* | 1.01 | 0.84, 1.20 | **0.73**[**] | 0.58, 0.92 | **0.76**[*] | 0.61, 0.94 |
| **Division** | | | | | | |
| *Barishal* | 1.00 | — | 1.00 | — | 1.00 | — |
| *Chattogram* | **1.73**[**] | 1.20, 2.52 | **2.12**[**] | 1.32, 3.55 | **2.18**[**] | 1.36, 3.62 |
| *Dhaka* | 1.15 | 0.77, 1.72 | 1.36 | 0.83, 2.31 | 1.39 | 0.86, 2.33 |
| *Khulna* | **2.71**[***] | 1.89, 3.93 | **2.66**[***] | 1.62, 4.50 | **2.76**[***] | 1.70, 4.65 |
| *Mymensingh* | 1.19 | 0.77, 1.84 | 1.49 | 0.84, 2.71 | 1.53 | 0.86, 2.75 |
| *Rajshahi* | **1.62**[*] | 1.11, 2.39 | **1.71**[*] | 1.03, 2.91 | **1.71**[*] | 1.05, 2.90 |
| *Rangpur* | 1.16 | 0.79, 1.72 | 1.55 | 0.94, 2.64 | 1.58 | 0.96, 2.67 |
| *Sylhet* | 1.11 | 0.73, 1.70 | 1.56 | 0.84, 2.90 | 1.55 | 0.85, 2.85 |
| **Smartphone Usage** | | | | | | |
| *Doesn't have any* | 1.00 | — | 1.00 | — | 1.00 | — |
| *Not a Smartphone* | 1.09 | 0.89, 1.34 | 0.90 | 0.72, 1.12 | 0.94 | 0.75, 1.17 |
| *Have a Smartphone* | 1.16 | 0.93, 1.45 | **0.60**[**] | 0.42, 0.84 | **0.64**[**] | 0.46, 0.88 |
| **Internet Usage** | | | | | | |
| *Never* | 1.00 | — | 1.00 | — | 1.00 | — |
| *Uses Internet* | **1.58**[***] | 1.30, 1.91 | **1.91**[***] | 1.41, 2.59 | **2.04**[***] | 1.52, 2.75 |
| **Frequency of reading Newspaper/Magazine** | | | | | | |
| *Not at all* | 1.00 | — | 1.00 | — | – | |
| Watches sometimes/ regularly | 1.22 | 0.87, 1.70 | 1.20 | 0.81, 1.74 | | |
| **Frequency of listening to Radio** | | | | | | |
| *Not at all* | 1.00 | — | 1.00 | — | – | |
| Watches sometimes/ regularly | 1.48 | 0.94, 2.26 | 1.49 | 0.87, 2.46 | | |
| **Frequency of Watching Television** | | | | | | |
| *Not at all* | 1.00 | — | 1.00 | — | 1.00 | — |
| Watches sometimes/ regularly | **1.24**[*] | 1.04, 1.47 | 1.10 | 0.90, 1.35 | 1.08 | 0.88, 1.33 |
| **Decision Maker of Respondent's Healthcare** | | | | | | |
| *Independently* | 1.00 | — | 1.00 | — | 1.00 | — |

*(Continued)*

**Table 3.** (Continued)

| Model Specification | Unadjusted Model[1] | | Full Model[2] | | Bootstrapped Model[3] | |
|---|---|---|---|---|---|---|
| *Jointly* | **0.48**\*\*\* | 0.38, 0.60 | **0.61**\*\* | 0.44, 0.85 | **0.61**\*\* | 0.45, 0.83 |
| *Others* | 0.84 | 0.65, 1.07 | 0.96 | 0.69, 1.35 | 0.87 | 0.63, 1.20 |
| **Decision Maker of Large Household Purchase** | | | | | | |
| *Independently* | 1.00 | — | 1.00 | — | – | |
| *Jointly* | **0.50**\*\*\* | 0.38, 0.65 | 0.91 | 0.59, 1.40 | | |
| *Others* | **0.67**\*\* | 0.50, 0.89 | 0.69 | 0.44, 1.08 | | |
| **Decision Maker of Visiting to Family/Relative** | | | | | | |
| *Independently* | 1.00 | — | 1.00 | — | 1.00 | — |
| *Jointly* | **0.53**\*\*\* | 0.42, 0.68 | 0.95 | 0.64, 1.41 | 0.89 | 0.64, 1.26 |
| *Others* | 0.99 | 0.76, 1.29 | **1.54**\* | 1.01, 2.35 | **1.51**\*\* | 1.11, 1.89 |
| **Decision Maker of spending Husband/Partner's Earning** | | | | | | |
| *Independently* | 1.00 | — | 1.00 | — | 1.00 | — |
| *Jointly* | **0.56**\*\*\* | 0.42, 0.75 | 0.94 | 0.65, 1.37 | 0.91 | 0.64, 1.31 |
| *Others* | 0.83 | 0.62, 1.13 | 1.00 | 0.69, 1.45 | 0.90 | 0.64, 1.29 |

[1]Univariable Logistic Regression Model,

[2]All Independent Variables are included into Fixed effect Multivariable Univariate Logistic Regression Model,

[3]Only Selected Variables are included into Fixed effect Multivariable Univariate Logistic Regression Model (Variables are selected by Backward Stepwise Selection Method and Estimates are calculated by bootstrapping),

[4] \*$p < 0.05$; \*\*$p < 0.01$; \*\*\*$p < 0.001$, [5] OR = Odds Ratio, CI = Confidence Interval.

secondary level only; from 2023 which includes frameworks, mandatory textbooks, and guides for teachers on mental health and wellbeing [48,49]. School-based curriculum containing mental health literacy in Tanzania, mental health package by WHO in Qatar, implementation of Canadian school mental health curriculum as guide in Malawi, "*Mental Health & High School Curriculum Guide*" ("*The Guide*") in Vietnam and Cambodia have shown significant improvement in keeping positive mental health attitude and wellbeing for both teachers and students [50–53].

60% of the women in our study are housewives and they were less likely to seek help in comparison with working women (working for any family members/Self-employed or working for others). Several studies have concluded that being a housewife has positive correlation with the negative mental health outcomes of women. Feeling overwhelmed and alone, they face rising anxiety which is driven by burnout, codependency, and how they see themselves [54,55]. Role of a housewife remains a deeply entrenched gender role, particularly in patriarchal societies, where women are expected to shoulder domestic responsibilities regardless of their employment status [56]. From an early age, societal norms condition women to adopt passive, emotional, obedient, and self-sacrificing behaviors—expectations that, once internalized, can profoundly diminish their sense of identity and self-worth within social interaction [57,58]. Study on housewives in India suggested that they were often pressured to prioritize the needs and expectations of others within their social environment [59]. Over time, this external focus can distort their sense of self, causing them to build a boundary within themselves, leading to a blurred identity and negative self-perception [60]. These reasons can be the explanation of why housewives in our study tend less seeking behavior. On the other hand, working women showed more concerned about their mental health, evidence from prior studies also support that working women had significantly better mental health [61–63]. We found significantly lower odds of help seeking in urban women. Sociologists contend that urban life often comes at the cost of meaningful human connection. Compared to residents of small towns or rural areas, individuals in large cities tend to experience weaker, less personal relationships and a decline in the overall quality of social interactions [64–66]. That results into urban dwellers are less likely than rural habitants to base their personal interactions on traditional links (such

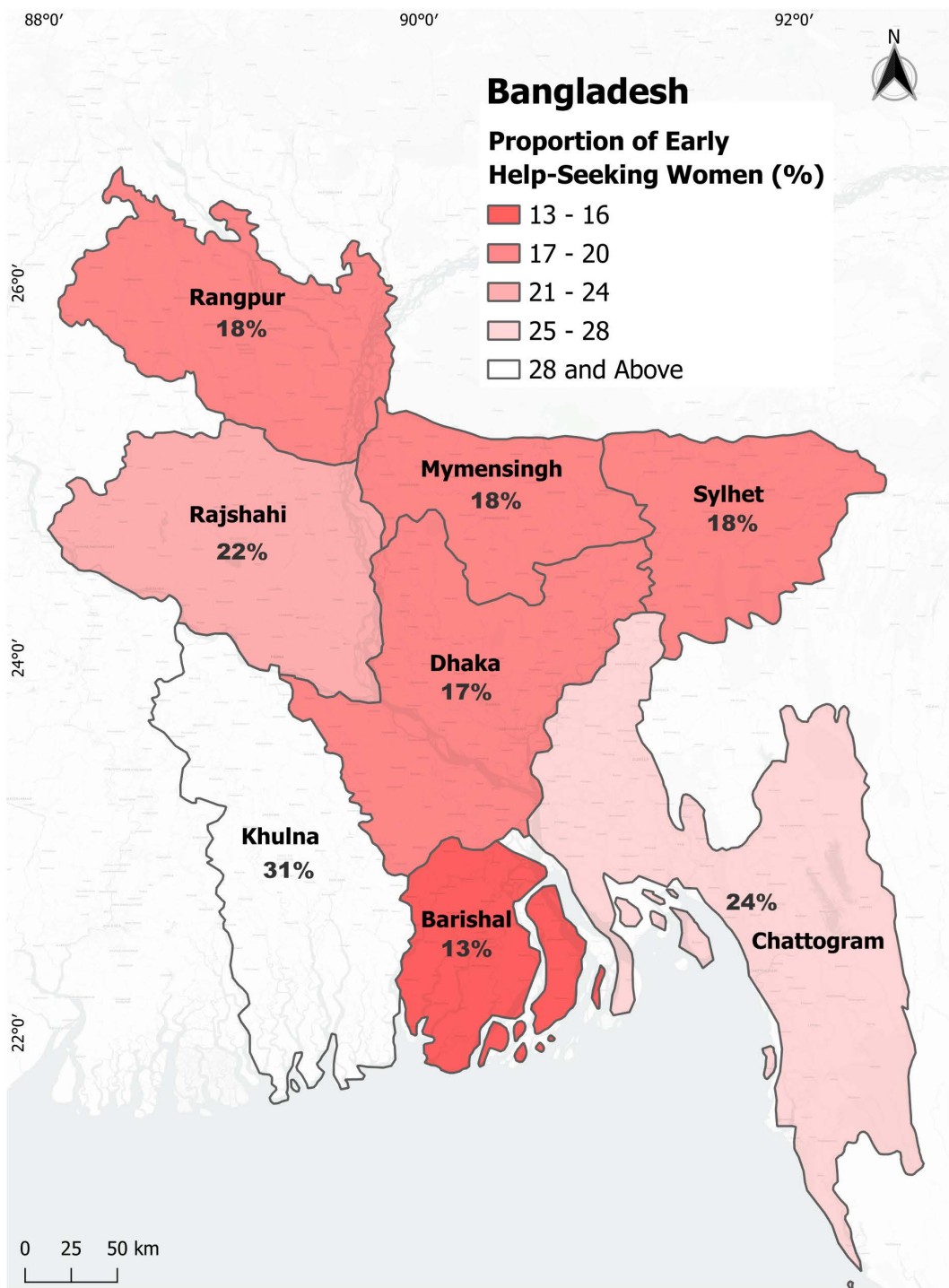

**Fig 3. Divisional distribution of the proportion of women who sought help for mild to moderate anxiety and depression.** Basemap: CartoDB Positron tiles © CARTO (CC BY 4.0), © OpenStreetMap contributors (ODbL, https://www.openstreetmap.org). Administrative boundary: geoBoundaries dataset (CC BY 4.0, www.geoboundaries.org).

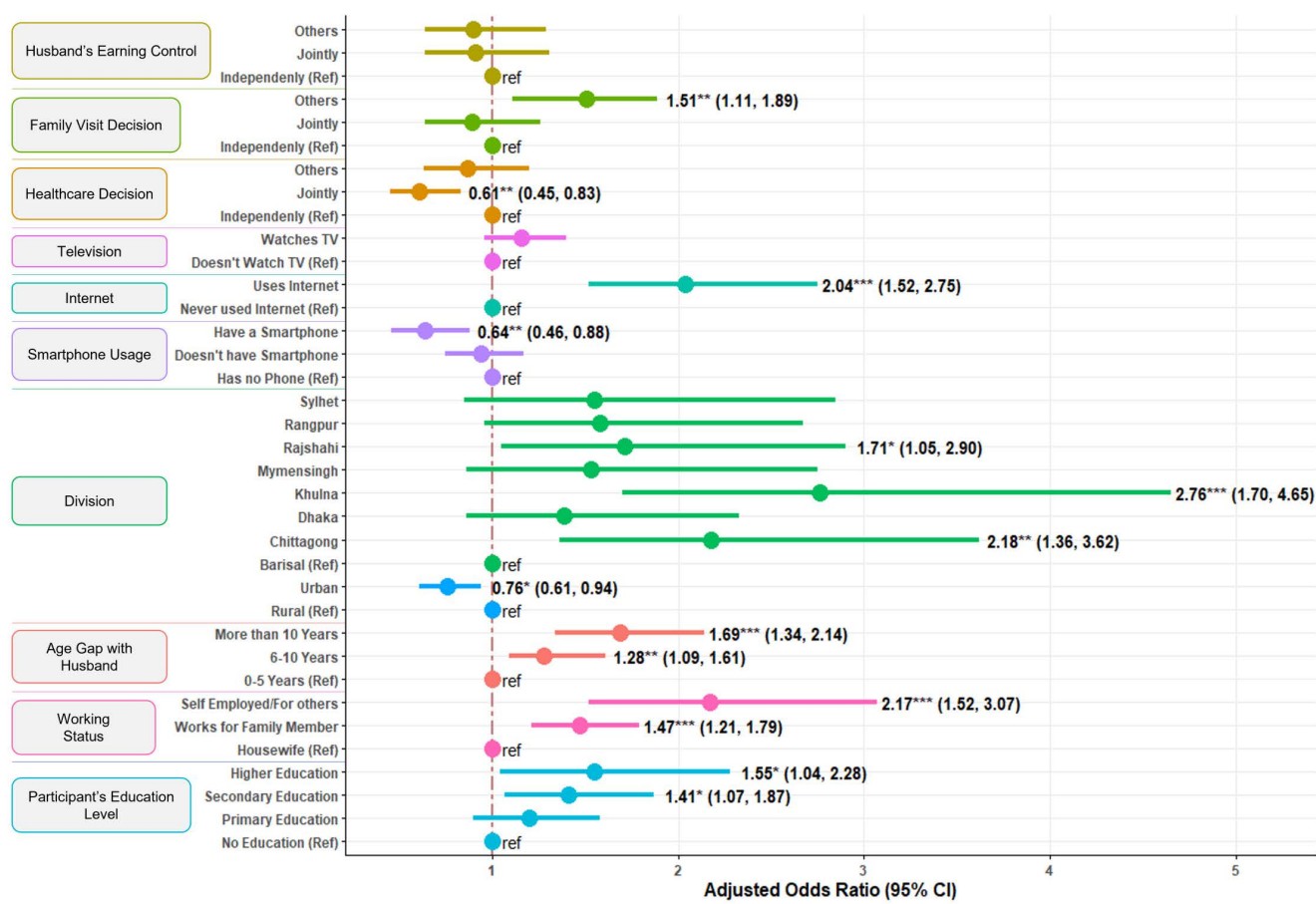

**Fig 4. Forest plot from showing the significant estimates from logistic regression model for help seeking in early stage.**

as family, relatives) [67]. Women from Khulna showed significantly higher tend of seeking help. We speculate this rise is because under the WHO initiative on urban governance for health and wellbeing, with leadership of the Mayor of Khulna City Corporation, a comprehensive four-year action plan has been formulated for the "Healthy City Khulna" initiative in 2020 [68]. The plan emphasizes the integration of health considerations into all policies, the promotion of healthy lifestyles including mental health issues [69].

Age difference with husband is founded to have significant impact on help seeking behavior of women in our study where older male-younger female marriages are the most common. This finding aligns with the finding from existing literature [70,71]. An Australian study indicated that couples with significant age differences tend to show lower resilience in managing challenges compared to couples who are closer in age [72]. This association reinforces a perspective prevalent in literature—that substantial age disparities in relationships are inherently negative to women's well-being, as women in such relationships often have reduced decision-making power within the household and face a higher risk of experiencing domestic turmoil [73,74]. Thus they look for help for their experiences more.

Autonomy for making decision on their own healthcare played a crucial role as we found respondent who had to make a decision jointly with her husband showed lower odds of seeking help compared to who made the decision independently. Across various healthcare domains—including accessing care, utilizing services, and choosing treatment options—women's autonomy is widely recognized as a critical component in effective decision-making [75]. In Pakistan,

cultural norms often discourage women from making independent decisions, leading to reduced autonomy [76]. Additionally, other studies have highlighted that traditional expectations regarding women's roles within the family vary across cultures, with many emphasizing the obligation for women have the right to share their opinion but must need to defer to their husbands in final decision-making processes [77–79]. Similar result was also found in our study that 85% of women in Bangladesh doesn't have the sole autonomy in their own healthcare decision. Previous research has indicated that empowering women to make autonomous decisions regarding their healthcare is associated with improved mental health outcomes, particularly in reducing levels of anxiety and lack of autonomy is linked to adverse outcomes, including poor mental health, reduced utilization of healthcare services, increased risk of malnutrition, and unmet family needs [80,81]. This explains why women who didn't have sole autonomy in decision making for their healthcare exhibited lower tendency to seek help for their mental health.

We put special focus on the role of media exposure on help seeking behavior, as previously in Bangladesh, the media campaigns (like television advertisements, billboards, leaflets, Radio shows) have successfully brought revolution in creating public awareness about childhood Vitamin A Supplements, immunization, smoking behavior, non-communicable diseases and health care seeking behavior regarding stigmatized issues such as tuberculosis and AIDS [82–87]. We found that the women who had access to the internet were more inclined towards seeking help while women owning a smartphone had a lower tendency to seek care when they faced the earlier symptoms of anxiety or depression. Stigma still looms large on women seeking mental healthcare from any facilities; many women often hide their symptoms to avoid getting shamed and judged [39]. Therefore, we assume access to the internet allows women to seek assistance for their mental health issues while ensuring privacy, convenience, and immediate access. Access to telehealth and online apps through the internet makes it easier in low-income settings or rural areas and among vulnerable populations to reach out to the help they need [88,89]. Women can share their experiences with professional caregivers or sometimes anonymously on different platforms while maintaining better control over their approach to finding help [89,90]. Moreover, mental health literacy (MHL) has been found to be higher among the participants using the internet, and increased professional help-seeking has been observed to be pursued by individuals with higher MHL, highlighting the significant relationship between mental health literacy and help-seeking behaviors [91,92]. This can also play as a contributing factor in higher mental health seeking tendencies among our participants during their early symptoms. All this evidence is suggestive of the fact that women with internet access are more aware and concerned, and have easy access to seek mental health care.

While an interesting finding has been observed in our study, that women with smartphones were less interested in seeking any help for their anxiety and depression. Report from a systematic review aligns with our findings that that 87% of its participants owned a mobile phone, but only 23% of them chose to use them as a means of asking for help [93]. The barriers that cause a lower number of help seekers include operational complexity and user engagement [94]. Furthermore, it has been reported that using a mobile phone has an initial 'blanket effect' that helps as an avoidance coping mechanism in the short run to moderately disengage from stressful and unpleasant emotional experiences [95–97]. Another systematic review reported that many individuals will not even communicate with mental health professionals in person and prefer contacting via smartphones, indicating that smartphone actually helps people to seek care for those who will not even consider seeking care in person [98].

Despite its widespread reach, television exposure showed no significant association with help-seeking behavior in our analysis. This reveals a critical, yet very underutilized opportunity in the Bangladeshi context. Television remains a powerful and cost-effective medium for influencing health behaviors, with its reach and popularity offering immense potential for public health promotion [29,99,83]. Evidence showed that women who watched TV dramas were significantly more likely to visit health facilities and use modern contraceptives, highlighting the medium's effectiveness—even within patriarchal and gender-constrained settings [30]. Despite the proven success of television in transforming several public health behaviors in Bangladesh, mental health remains absent from the outdated National Broadcast Policy (2014) of

Bangladesh [100]. This situation is not just due to a missed opportunity; it's due to policy gap as well. As mental health problems like anxiety and depression are becoming more common, especially among women & young people, this indicates a serious failure to use a media for public health promotion.

In general, people ignore the early signs and symptoms of illness, often avoiding professional help and instead relying on personal beliefs, cultural norms, or stigma-driven assumptions and often socially determined [101,102] In Bangladesh, this situation is more prominent [103,104]. Health conditions are frequently misunderstood or minimized due to low health literacy, widespread stigma, and a tendency to attribute illness to fate, spiritual causes, or bad luck [105–110]. Many individuals opt for self-treatment, traditional healers, or home remedies rather than visiting a healthcare provider, partly due to financial constraints, mistrust in the healthcare system, or a deep-rooted belief that symptoms will resolve on their own [111–113]. This also results into overall low proportion of help seeking behavior in case of mental health as well.

## Conclusion

This study enlightens a critical yet neglected side, the alarmingly low levels of help-seeking behavior among women experiencing early-stage anxiety and depression in Bangladesh, revealing a pattern influenced by their working status, education level, autonomy, and media exposure. Housewives are among the most vulnerable groups to mental distress in Bangladesh, yet they are also the least likely to seek help—a critical reality that demands urgent, multisectoral action. Among all factors, Media Exposure is one of the most powerful factors, but Television remains as a underutilized tool for mental health transformation till now. It remains absent in the country's mental health strategy, and is yet to be mobilized for mental health awareness and promotion programs. This omission represents not just a missed opportunity, but a serious blind spot.

## Recommendations

### 1. Leverage television for mass mental health promotion

Given television's deep penetration in grassroot level of the society, national mental health promotion campaigns using TV dramas, public service announcements, and entertainment-education formats should be urgently prioritized as well with the strategies that have been undertaken to utilize smartphone-, internet-based mental health service and promotion. Mental health literacy can be embedded within popular and culturally relatable programming to reduce stigma, encourage early help-seeking, and normalize conversations about mental well-being in underdeveloped regions where scarcity of modern technology such as (Smartphone, Internet) is inaccessible. Evaluating the effectiveness of currently aired-programs should be assessed thoroughly to ensure that mass media serves its intended educational and social functions. Continuous assessment and adaptation are also necessary to enhance the impact of these programs to make these more relevant and beneficial to the target population.

### 2. Train housewives as peer counselors

Evidence from diverse low-resource settings has demonstrated that peer counseling by laywomen can effectively improve health behaviors and psychosocial outcomes, particularly when rooted in shared lived experiences and community trust [114–117]. Training housewives as peer counselors can capitalize on existing community trust and shared lived experiences to offer emotional first aid. This approach should help reduce shame, build trust, and encourage women to seek support for mental health issues in a safe and familiar environment. Feasibility of such intervention needs urgent exploration. Such peer-based interventions can be particularly effective in contexts where formal mental health services are scarce, making it both a scalable and culturally sensitive strategy according to the needs of communities.

### 3. Promote shared domestic responsibilities and autonomy-supportive policies

Empirical evidence consistently indicates that equitable distribution of household and cognitive tasks, alongside supportive policies to recognize and enable women's autonomy, leads to improved decision-making power, psychological well-being,

and health-seeking behaviors among housewives [118–121]. Shared domestic responsibilities and support policies that give housewives more autonomy, time for self-care, and recognition beyond their household roles should be heavily promoted to turn housewives' unrecognized labor into visible power and give them the respect they deserve. By shifting the narrative from "household duty" to "shared responsibility", a culture can be created where women's contributions, inside and outside the home, are celebrated as integral to societal well-being and development.

### 4. Thoroughly integrate mental health education into the national curriculum

Embedding comprehensive mental health and well-being education into the national school curriculum; from early grades through higher education; can help normalize mental health discussions from a young age. This initiative should aim to build resilience, emotional literacy, and help-seeking behaviors before individuals transition into domestic or workforce responsibilities. While this requires substantial coordination between education and health sectors, its long-term impact on generational attitudes toward mental health is considerable.

### 5. Support through policy and cross-sector collaboration

To ensure the longevity and structural integration of these initiatives, robust government policies must provide institutional support. Partnerships between government, NGOs, public and private sectors can enhance reach and innovation, while collaboration with mental health professionals ensures the ethical and effective delivery of interventions. Such cross-sectoral alignment is critical to embedding these recommendations within a sustainable, national mental health strategy.

## Future scope

Future research should be strategically embedded within the broader vision of a national mental health strategy that prioritizes scalable, culturally relevant, and community-anchored solutions. A key area of investigation involves assessing the feasibility of developing television-based mental health interventions such as awareness campaigns, entertainment-education formats, and public service announcements that are culturally tailored to reduce stigma and promote early help-seeking behaviors. Such interventions must be rigorously evaluated using experimental designs, or where feasible, natural experimental approaches to measure both effectiveness and contextual fit. Full-scale trials should also assess long-term behavioral impact, cost-effectiveness, and potential for national scalability. Importantly, this line of research should not be pursued in isolation. It must be complemented by parallel efforts to train lay peer counselors, reform educational curricula, and advocate for gender-equitable support policies, all contributing to a cohesive, multisectoral national mental health framework. These research endeavors can also inform similar strategies in other low- and middle-income countries (LMICs), positioning Bangladesh as a model for localized, media-driven, self-supported mental health help delivery and promotion.

## Supporting information

**S1 Fig. Assumption Assessment of the Initial Model before Bootstrapping.**
(TIF)

**S1 File. Supporting Information (Analysis Pathway, and Interpolation).**
(DOCX)

**S2 Fig. Cluster-wise Proportion of Women Sought for Help in Early Stage of Depression and Anxiety.**
(TIF)

## Author contributions

**Conceptualization:** Tanvir Ahmed, Salma Tasnim Luthfa, Amatul Haque Chaahat.

**Data curation:** Tanvir Ahmed.

**Formal analysis:** Tanvir Ahmed.

**Investigation:** Tanvir Ahmed.

**Methodology:** Tanvir Ahmed, Salma Tasnim Luthfa, Amatul Haque Chaahat, Azaz Bin Sharif.

**Software:** Tanvir Ahmed.

**Supervision:** Azaz Bin Sharif.

**Validation:** Tanvir Ahmed, Masudur Rahman Kanchon, Azaz Bin Sharif.

**Visualization:** Tanvir Ahmed.

**Writing – original draft:** Tanvir Ahmed, Salma Tasnim Luthfa, Amatul Haque Chaahat.

**Writing – review & editing:** Tanvir Ahmed, Masudur Rahman Kanchon, Azaz Bin Sharif.

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
