## [Decision Letter · Decision Letter 0]

5 Aug 2025

PMEN-D-25-00281

Determinants of Early Help-Seeking Behavior for Mental Distress Among Women in Bangladesh: A Bootstrapped Logistic Regression Analysis of Nationally Representative Survey Data

PLOS Mental Health

Dear Dr. Ahmed,

Thank you for submitting your manuscript to PLOS Mental Health. After careful consideration, we feel that it has merit but does not fully meet PLOS Mental Health’s publication criteria as it currently stands. Therefore, we invite you to submit a revised version of the manuscript that addresses the points raised during the review process.

We look forward to receiving your revised manuscript.

Kind regards,

Lambert Zixin Li, Ph.D.

Academic Editor

PLOS Mental Health

Journal Requirements:

https://journals.plos.org/mentalhealth/s/figures

https://journals.plos.org/mentalhealth/s/figures#loc-file-requirements

2. We have amended your Competing Interest statement to comply with journal style. We kindly ask that you double check the statement and let us know if anything is incorrect.

4. Thank you for uploading your study's underlying data set. Unfortunately, the repository you have noted in your Data Availability statement does not qualify as an acceptable data repository according to PLOS's standards.

5. Some material included in your submission may be copyrighted. According to PLOS’s copyright policy, authors who use figures or other material (e.g., graphics, clipart, maps) from another author or copyright holder must demonstrate or obtain permission to publish this material under the Creative Commons Attribution 4.0 International (CC BY 4.0) License used by PLOS journals. Please closely review the details of PLOS’s copyright requirements here: PLOS Licenses and Copyright. If you need to request permissions from a copyright holder, you may use PLOS's Copyright Content Permission form.

Potential Copyright Issues: Figure 3: please (a) provide a direct link to the base layer of the map (i.e., the country or region border shape) and ensure this is also included in the figure legend; and (b) provide a link to the terms of use / license information for the base layer image or shapefile. We cannot publish proprietary or copyrighted maps (e.g. Google Maps, Mapquest) and the terms of use for your map base layer must be compatible with our CC-BY 4.0 license.

Additional Editor Comments (if provided):

Dear Authors,

Thank you for submitting your manuscript to PLOS Mental Health. After review, we invite you to submit a revised version addressing the comments provided by the reviewers.

While your study shows potential, the reviewers have identified several areas that require clarification or improvement. Please submit a point-by-point response detailing how each comment has been addressed, along with a revised version of the manuscript.

We look forward to receiving your revision.

Sincerely,

Lambert Zixin Li, PhD

PLOS Mental Health

Reviewers' comments:

Reviewer's Responses to Questions

**Comments to the Author**

1. Does this manuscript meet PLOS Mental Health’s publication criteria ? Is the manuscript technically sound, and do the data support the conclusions? The manuscript must describe methodologically and ethically rigorous research with conclusions that are appropriately drawn based on the data presented.

Reviewer #1: Yes

Reviewer #2: Yes

2. Has the statistical analysis been performed appropriately and rigorously?

Reviewer #1: Yes

Reviewer #2: Yes

3. Have the authors made all data underlying the findings in their manuscript fully available (please refer to the Data Availability Statement at the start of the manuscript PDF file)?

Reviewer #1: Yes

Reviewer #2: Yes

4. Is the manuscript presented in an intelligible fashion and written in standard English?

Reviewer #1: Yes

Reviewer #2: Yes

5. Review Comments to the Author

Reviewer #1: Timely and policy-relevant topic in global mental health, particularly among women in LMICs.

To strengthen this manuscript further:

1. Title is accurate but long. Suggest trimming for clarity “Determinants of Early Mental Health Help-Seeking Among Women in Bangladesh: A Nationally Representative Analysis”

2. Abstract: Results can be grouped more concisely, too many aORs may overwhelm readers.

3. Introduction:

a. Some sentences are long and could be broken into simpler, clearer statements (e.g., lines 54-57, 281-283 and 396-399).

b. Consider avoiding repetition (e.g., lines 48–57 and 63–66 discuss the burden and importance of early intervention similarly).

4. Methods:

a. Clarify “bootstrapped fixed-effects logistic regression” for general readers. Are the strata treated as fixed intercepts?

b. Variable selection process ("backward stepwise") is clear but could benefit from a brief note on why it was preferred over LASSO or AIC-based selection.

c. Although the manuscript refers to several figures, they are not embedded in the main text. While the figures have been provided as supplementary files, their resolution is too low to interpret the content effectively.

5. Results:

a. Clarify the contrast between “Smartphone users” vs. “Internet users”. This is nuanced but may confuse some readers.

b. Consider summarizing key findings in a sentence or two before launching into detailed odds ratios.

c. Group related predictors (e.g., all autonomy-related ones) to reduce the density of Table 03.

6. Discussion: Some sections lean toward advocacy without adequate empirical support. For instance, the suggestion to train housewives as peer counselors (lines 412–414) is conceptually valuable but would benefit from supporting evidence or citations. Similar concerns apply to the strong recommendations presented in lines 399–403 and 414–416, which would be more compelling if grounded in existing research or programmatic examples.

7. Policy and practice implications:

a. Consider prioritizing recommendations in order of impact and feasibility.

b. Frame future research as part of a broader national mental health strategy.

8. Capitalization has been used unnecessarily within sentences, for example, in phrases like “Strata,” “Smartphone,” “Internet,” “Newspaper/Magazine,” “Radio,” “Television,” “Age,” “Education Level,” “Occupation,” “Marital Status,” “Religion,” and “Media Exposure”, which disrupts consistency and proper grammar.

Reviewer #2: This is a well designed study that promotes gender equity and inclusion in design of mental health programmes in resource poor settings. It was observed that 92.4% of respondents were in a particular religious inclination and therefore it may be difficult to analyse the influence of religion on the care seeking behaviour of these women. In subsequent studies, the authors may wish to consider exploring the role of religion on their study outcomes or possibly compare their findings with those of women from other socio-cultural (religious) settings. This is an important factor that the authors may wish to consider.

6. PLOS authors have the option to publish the peer review history of their article (what does this mean? ). If published, this will include your full peer review and any attached files.

**Do you want your identity to be public for this peer review?** For information about this choice, including consent withdrawal, please see our Privacy Policy .

Reviewer #1: No

Reviewer #2: No

---

## [Editor Report · Decision Letter 1]

20 Aug 2025

Determinants of Early Mental Health Help-Seeking Among Women in Bangladesh: A Nationally Representative Bootstrapped Regression Analysis

PMEN-D-25-00281R1

Dear Dr Ahmed,

We are pleased to inform you that your manuscript 'Determinants of Early Mental Health Help-Seeking Among Women in Bangladesh: A Nationally Representative Bootstrapped Regression Analysis' has been provisionally accepted for publication in PLOS Mental Health.

Best regards,

Lambert Zixin Li, Ph.D.

Academic Editor

PLOS Mental Health

Thank you for thoroughly addressing all reviewer comments. I have carefully reviewed the revised manuscript, your response letter, and the highlighted changes. I find that the paper is of publishable quality.